# Preparation and Properties of Pea Starch/ε-Polylysine Composite Films

**DOI:** 10.3390/ma15062327

**Published:** 2022-03-21

**Authors:** Zuolong Yu, Deping Gong, Chao Han, Yunxiao Wei, Changchun Fu, Xuejiao Xu, Youri Lu

**Affiliations:** 1Biology and Environmental Engineering College, Zhejiang Shuren University, Hangzhou 310015, China; chaohan96@hotmail.com (C.H.); lvwyx@163.com (Y.W.); 17816184576@163.com (C.F.); xuxuejiao@zjsru.edu.cn (X.X.); lu_12027@126.com (Y.L.); 2School of Innovation & Entrepreneurship, Zhejiang Shuren University, Hangzhou 310015, China; gongdeping@zjsru.edu.cn

**Keywords:** pea starch, ε-poly lysine, composite film, antibacterial, properties

## Abstract

The composite films comprising pea starch (St) and ε-polylysine (PL) as the matrix and glycerol and sodium alginate as the plasticizers were investigated. The rheological properties, mechanical properties, Fourier transformed infrared spectroscopy, water vapor permeability (WVP), oil permeability, microstructure, thermogravimetry (TGA), and antimicrobial properties of the composite films were analyzed. The properties of the composite films with different mass ratios of St/PL varied significantly. First, the five film solutions were different pseudoplastic fluids. Additionally, as the mass ratio of PL increased, the tensile strength of the blends decreased from 9.49 to 0.14 MPa, the fracture elongation increased from 38.41 to 174.03%, the WVP increased, and the oil resistance decreased substantially. The films with a broad range of St/PL ratios were highly soluble; however, the solubility of the film with a St/PL ratio of 2:8 was reduced. Lastly, the inhibition of *E. coli*, *B.subtilis*, and yeast by the films increased with increasing mass ratios of PL, and the inhibition of *B.subtilis* was the strongest.

## 1. Introduction

Edible packages and films are in line with the strategy of green development and life and health maintenance because they use natural polymers, such as polysaccharides, proteins, and lipids, as the substrate for food packaging and coating [1,2,3,4]. The development of starch-based edible packaging films has been extensive since the 1980s [5,6]. For example, starch-based edible packages and films from different sources are found to have different mechanical properties, rheological properties, and gas permeability due to their different proportions of straight and branched chains and varying crystallinity [7,8]. In addition, physical and chemical modifications of starch can change its gelatinization temperature and viscosity, improving the thermal stability, mechanical properties, and water resistance of a film [9,10,11]. Therefore, the elasticity, flexibility, and processability of an edible film can be strengthened by adding different components to its formula [12,13].

The properties of starch blended with other substrates can be designed directionally to provide a theoretical basis for intelligent applications. For example, increasing the polyvinyl alcohol (PVA) content in a blend of starch and PVA elevates the mechanical properties and barrier properties of the film significantly but decreases the permeability significantly [14]. In addition, edible films of acetylated cassava starch and pea protein isolate are produced using conventional blown-film extrusion; when the proportion of pea protein isolate reaches 20%, it has good processability, and protein improves the oxidative stability and structural integrity of edible packages [15]. Moreover, the blending of starch with different types of polyethylene can change the rheology, thermal stability, and mechanical properties of the blends to varying degrees, facilitating the quantitative application of starch in materials [16,17,18].

On the other hand, ε-polylysine (PL), containing many ammonia residues, has antibacterial activities, high water solubility, high safety, and thermal stability in food anti-corrosion, weight loss health care, and drug carriers and has a market value in other aspects [19,20,21,22]. PL is also used as a component of packaging films; for example, PL and chitosan blends exert the bacteriostatic effect of both substances [23]. In addition, adding PL to γ-polyglutamic acid to form envelope material can effectively protect the normal physiological activities of probiotics [24]. Moreover, using PL and natural extracts to coat ready-to-eat foods can effectively prolong the shelf life of the foods [25,26]. Lastly, adding different percentages of PL to starch film can effectively inhibit microbial growth [27].

In this work, the effects of blends with different proportions of pea starch (St) and PL on film-forming performance and microbial inhibition were investigated; the result of this study will provide a data basis for the application of functional packaging films.

## 2. Materials and Methods

### 2.1. Preparation of Composite Films

We mixed St (Yantai Oriental Protein Tech Co., Ltd., Yantai, China) and PL (average number-average molar mass 1.5 kDa; Nanjing Shineking Biotech Co., Ltd., Nanjing, China) at mass St/PL ratios of 10:0, 8:2, 6:4, 4:6, or 2:8 to create 10-g blends. First, St was added into a three-necked flask with 70 mL of water. Then, 0.6 g of sodium alginate (Shanghai Aladdin Biochemical Technology Co., Ltd., Shanghai, China) and 1.2 g of glycerol (Shanghai Aladdin Biochemical Technology Co., Ltd., Shanghai, China) were added. After the mixture was fully dispersed, it was heated for gelatinization in an 85 °C water bath and stirred at 300 rpm for 30 min. After gelatinization was completed, PL was dissolved into 30 mL of water, adjusted to the pH to 8, added into the three-necked bottle, and stirred for 20 min to prepare the composite film solution. Next, the films were cast on a plastic plate and dried in the oven at 55 °C. After the films were covered for 24 h, their properties were analyzed.

### 2.2. Measurement of Rheological Properties

We measured the apparent viscosity shear rate curves of the five composite film solutions using a rheometer (MCR 102, Anton-Paar, Shanghai, China). Measurements were conducted at 85 °C with a cone plate with a diameter of 50 mm, 1°. In addition, the measurement mode was flow curve. With the shear rate as the variable, the range was 0.01–100 s^−1^, the mode was linear scanning, and 50 variable points were selected.

### 2.3. Measurement of Mechanical Properties

We determined the thickness of each composite film as the average value of five spots on a film measured using a micrometer caliper (±0.01 mm). In addition, we cut the films into 100 × 5 mm^2^ rectangles for tensile testing. The tensile strength (T_S_) and fracture elongation (E) of the films were measured using a physical property tester (TA.XT Plus, Stable Micro System, London, Britain). Each group measured three transverse and longitudinal samples in the vertical direction, a total of six parallel samples, and calculated the average T_S_ and E. The spacing was 50 mm, and the sample speed was 100 mm/min. The formula was as follows:(1)$$T_{S}=F/S$$
where T_S_ was the tensile strength (MPa), F was the tensile force of the film (N), and S was the cross-sectional area of the film (m^2^).
(2)$$E=L-\frac{L_{0}}{L_{0}}\times 100\%$$
where E was the fracture elongation (%), L_0_ was the original standard distance of the sample (mm), and L was the standard distance of sample fracture (mm).

### 2.4. Fourier Transformed INFRARED spectroscopy

For Fourier transformed infrared spectroscopy (FTIR), 1 mg of a composite film sample combined with 100 mg of potassium bromide was ground in a mortar. Then, the ground mixture was compressed into a disk shape with a diameter of 13 mm and a thickness of 1 mm using a mold. Each sample was scanned from 4000 to 500 cm^−1^ using an FTIR spectrometer (FTIR-650, Bruker, Karlsruhe, Germany) to produce a spectrum at a resolution of 4.0 cm^−1^ over 64 scans.

### 2.5. Measurement of Water Vapor Permeability (WVP)

We measured the WVP of the composite films using the quasi-cup method [28,29]. In summary, we coated a composite film on a cell containing a certain amount of dry CaCl_2_ and sealed the cell with melted wax. Then, we placed the cell in an environment with 100% humidity at 25 °C. Next, we measured the weight of the cell every 24 h for 1 week. The WVP was defined as the amount of water vapor transported through the film per unit time, pressure, and film area:(3)WVP=Δm×d/(A×t×ΔP)
where WVP was the water permeability coefficient (g·mm/m^2^·d·KPa), Δm was the increase in the cell weight (g), d was the thickness of the film (mm), A was the effective area of the film (m^2^), t was the time interval of the measurement (d), and ΔP was the vapor pressure difference on both sides of the sample (KPa).

### 2.6. Measurement of Oil Permeability

We measured the oil permeability of the composite film according to a previously established procedure [28] with a minor modification. First, we sealed a tube containing 5 mL of peanut oil with a composite film. Then, we inverted the tube onto a filter paper. Next, we measured the weight of the filter paper for 1 week to check the oil permeability coefficient (OPC) of the composite film. The OPC was defined through the following equation:(4)OCP=Δm×d/A×T
where OPC was the oil permeability coefficient (g·m/m^2^·d), Δm was the weight increase of the filter paper (g), d was the thickness of the composite film (mm), A was the effective area of the composite film (m^2^), and T was the time taken for oil permeation (days).

### 2.7. Measurement of Thermogravimetry

The thermal stability of the powder samples was analyzed using the TGA and differential thermogravimetric analysis (DTG) with a thermal analyzer (DTG-60, Shimadzu, Kyoto, Japan). The TGA experiments were conducted at a heating rate of 10 °C/min from room temperature to 600 °C under constant purging, with nitrogen gas at a flowing rate of 2.5 mL/min.

### 2.8. SEM

The morphology of the composite films was investigated using a scanning electron microscope (S-570, Hitachi, Tokyo, Japan). At a working voltage of 10 kV, the fractured surfaces of the notched Izod impact samples in liquid nitrogen were observed. Before viewing, the observed surfaces were coated with gold, and the observation sites were located in the central regions of the surfaces.

### 2.9. Antibacterial Assays

We chose *Escherichia coli* (ATCC 25312), *Bacillus subtilis* (ATCC 23857), and yeast (*Saccharomyces cerevisiae*, ATCC 204508) (from the college’s microbiology laboratory) to test the antimicrobial properties of the composite films [23]. We put three prepared media, routine broth, Luria–Bertani broth, and yeast extract peptone dextrose, into triangular bottles and wrapped up the Petri dishes, coating sticks, and other items to sterilize with damp heat. Then, a medium was poured into a plate on an ultra-clean workbench, marked, and cooled. Next, 10 μL (10^7–8^ CFU) of a bacterial suspension was added into the corresponding culture medium, coated evenly, and allowed to infiltrate. Afterward, the films prepared with different St/PL ratios were made into small discs with a diameter of about 10 mm with a hole punch and carefully placed on a bacteria-coated plate to incubate in a constant temperature incubator at 37 °C for 24–48 h. Subsequently, the sizes of the antimicrobial circles on the plates were observed to assess the antimicrobial activity of the films.

## 3. Results and Discussion

### 3.1. Rheological Properties of the Composite Films

The rheological properties of the films with different St/PL mass ratios were compared (Figure 1). With the increase in shear rate, the viscosity of the five samples first rose to the highest point and then decreased steadily. The films were pseudoplastic fluids in the case of shear thinning [30]. The viscosity was highest for the sample with a St/PL ratio of 4:6, at 1.03 × 106 MPa·s, followed by the films with the ratio of 2:8, 10:0, 6:4, and 2:8. In addition, the shear rate of the 4:6 film was 0.241 s^−1^. After entering the steady state, the viscosity of the film solution with a St/PL ratio of 10:0 reached the maximum, while the other samples decreased with increasing proportions of PL, mainly due to the decrease of viscosity accompanied by the increasing proportions of small molecules (PL) [31]. However, the solubility of PL in the formula system and St would lead to inconsistency between the viscosity of the sample and the formula; thus, the 4:6 film had the highest viscosity, followed by the 6:4, 2:8, and 8:2 films.

### 3.2. Mechanic Properties of the Composite Films

The films with varying St/PL mass ratios had significantly different mechanical properties (Figure 2). The T_S_ of the composite films decreased, and the break elongation increased with increasing proportions of PL. When the St/PL mass ratio changed from 10:0 to 6:4, the T_S_ decreased significantly; however, the declining trend was slowed from 6:4 to 2:8 because the main component of the composite film changed from St to PL, and PL’s molecular weight was lower than St’s. The winding and rigid structure of starch macromolecules display great differences in macro mechanical properties. Although the amount of PL increased because its lower molecular weight reduced the rigidity of the network structure, the T_S_ of the composite film was less affected when the St/PL mass ratio exceeded 6:4. In addition, break elongation increased with the increasing proportion of PL because PL also played a plasticizing role in the system. PL increased the sliding between starch macromolecules, enhancing the break elongation value from 38.41 to 172.03%. However, a film could not be formed when PL was only used as the matrix and sodium alginate and glycerol were used as plasticizers. Therefore, in this work, there was no film with a St/PL mass ratio of 0:10 to be tested for performance.

### 3.3. FTIR Spectra of the Composite Films

The FTIR spectra of the films with different St/PL mass ratios were compared (Figure 3). The characteristic absorption peaks of PL were at 3382, 3362, 3081, 1633, and 1535 cm^−1^ [32]. Meanwhile, St had characteristic absorption peaks at 3360, 2928, 1646, and 1021 cm^−1^ [33]. The composite films had obvious absorption peaks at corresponding positions (Figure 3). The stretching vibration absorption peak of C-O-C, wide C-H stretching vibration band, free hydroxyl band (non-bonded -OH stretching vibration band), and hydrogen-bonded hydroxyl band were at 1100, 2850–3000, 3600–3650, and 3200–3570 cm^−1^, respectively. The molecular structure of glycerol contains primary alcohol and secondary alcohol. There were wide and strong stretching vibration peaks of polyconnective OH at 3287.3 cm^−1^ and symmetric and antisymmetric stretching vibration absorption peaks of C-O bond of glycerol at 1115.8–994 cm^−1^ [34]. Only hydrogen bonds between each substance indicated that no strong chemical reaction had taken place between each component. It was more favorable for PL to maintain antibacterial activity.

### 3.4. Gas Permeability of the Composite Films

The WVP and oil permeability of the composite films with different St/PL mass ratios were compared (Figure 4). The water vapor transmittance increased with the rising proportion of PL because of the large number of polar amino groups in PL, a hydrophilic cationic polymer. In addition, a simple mechanical stirring of a film solution will deteriorate the barrier performance of a composite membrane to water vapor [35].

Meanwhile, the oil permeability of the composite films increased with the increasing proportion of PL. The composite film with a St/PL ratio of 10:0 had the best oil resistance. In addition, the oil resistance of the films with a St/PL mass ratio of 4:6 and 2:8 was substantially weakened. With the increasing proportions of PL, the films formed a stable hydrogen bond into the increase of the oil-philic group, and the oil entered the composite films to destroy their hydrogen bond and structure. As a result, the evacuation degree of the composite films was greatly improved, decreasing the barrier property considerably.

### 3.5. Thermogravimetric Analysis of the Composite Films

According to thermogravimetric analysis (Figure 5a), all samples had a water evaporation weight loss stage at 100–160 °C. With the increasing proportions of PL, the starting time of thermal weight loss in the second stage also gradually advanced (Figure 5b). Because PL was well-integrated into the starch molecules, it resulted in their loose composition and early thermal decomposition. The samples with St/PL ratios of 8:2, 6:4, and 4:6 did not form an obvious slope, indicating that St and PL molecules were evenly mixed, intertwined, and gradually thermally degraded. When the St/PL mass ratio was 2:8, excessive PL hindered the mixing of various substances; as a result, the sample displayed four thermogravimetric stages, which were not conducive to the film’s performance.

### 3.6. SEM of the Composite Films

The cross-sections of the composite films were obtained during the scanning electron microscopic analysis (Figure 6). After gelatinization, the crystal region of starch was destroyed and was well soluble with other substances. The internal dispersion of each sample was uniform, and the sections were smooth. However, with the increasing proportion of PL, the gel particles in the cross-sections gradually increased. Because PL molecules acted as the matrix, changing the force of the secondary bonds between different substances, the starch molecules were dispersed into PL, resulting in the formation of colloidal particles.

### 3.7. Antimicrobial Activities of the Composite Films

As the proportion of PL in the films increased, the inhibition of the two bacteria and one fungus by the composite films was gradually enhanced (Figure 7 and Table 1). The antibacterial mechanism of PL has been reported [36,37,38]. PL can act on the cell wall, cell membrane, genetic material, and functional proteins, leading to cell decay. However, when PL is mixed with other substances to make products, its functions vary depending on other substances in the mixture and the method used. The results indicated that the inhibitory effect of PL on *B.subtilis* was stronger than that of *E. coli* and yeast. When the concentration of PL reached 20%, the composite film did not inhibit *E. coli* or yeast. More detailed experiments are needed to study the minimum inhibitory concentration of PL in starch films.

## 4. Conclusions

Five St/PL blends prepared by casting had different properties. As the mass ratio of PL increased, the rheological behavior of the film solution (St/PL: 6:4, 4:6, and 2:8) increased and decreased sharply at a shear rate of 0.241 s^−1^. Meanwhile, the Ts and E of the composite films declined from 9.49 to 0.14 MPa and increased from 38.41% to 174.03%, respectively. The water vapor permeability and oil permeability increased regularly. Only when the St/PL mass ratio was 2:8 did the oil permeability of the composite film have no barrier ability. In addition, FTIR analyses revealed that St and PL had secondary bonds; thermogravimetry revealed the advance of a thermal degradation peak; and SEM revealed gel formation. All these findings revealed that St and PL were miscible in a certain proportion. Due to the antimicrobial properties of PL, they can be greatly reduced after the preparation of the starch hybrid film. Lastly, the films exerted different inhibitory effects on *E. coli*, *B. subtilis*, and yeast.

## Figures and Tables

**Figure 1 materials-15-02327-f001:**
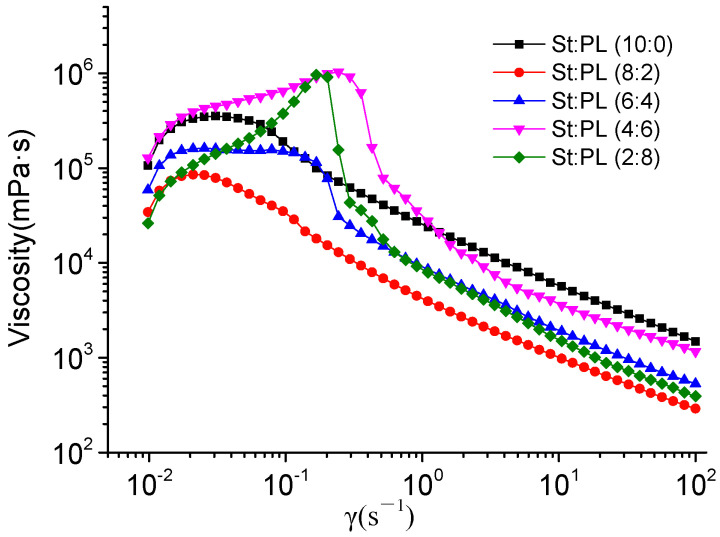
Rheological curves of the film solutions with different St/PL mass ratios.

**Figure 2 materials-15-02327-f002:**
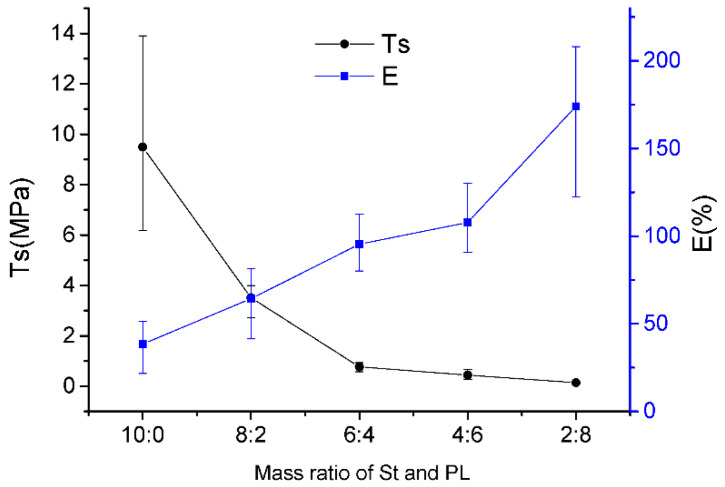
Mechanic curves of the composite films with different St/PL mass ratios.

**Figure 3 materials-15-02327-f003:**
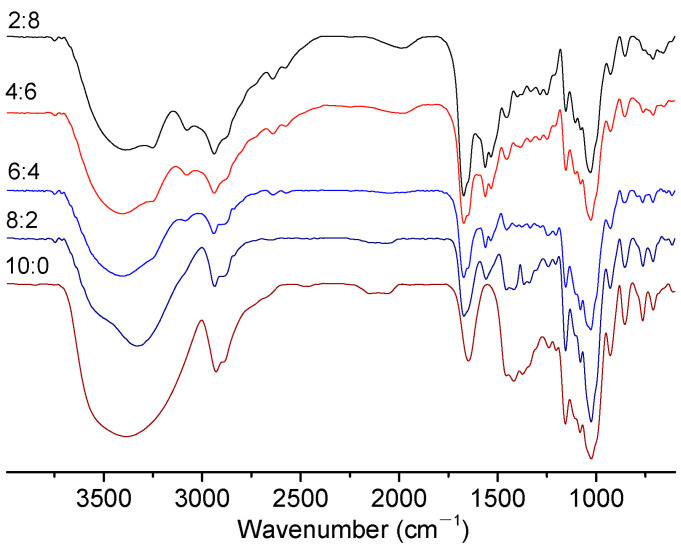
FTIR spectra of the composite films with different St/PL mass ratios.

**Figure 4 materials-15-02327-f004:**
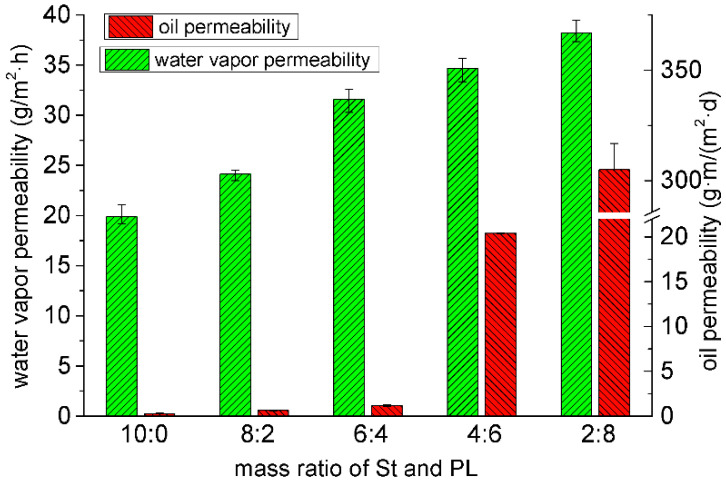
Water vapor permeability and oil permeability of the composite films with St/PL mass ratios.

**Figure 5 materials-15-02327-f005:**
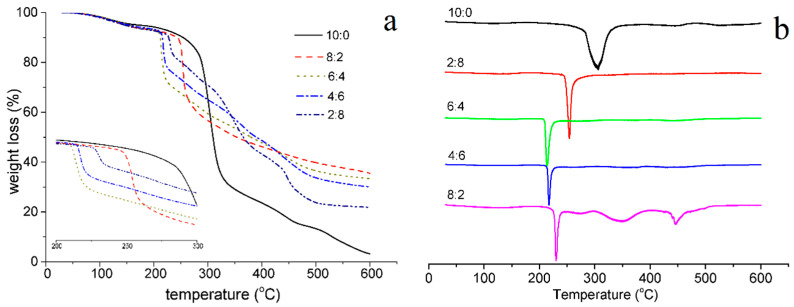
TGA and DTA curves of the composite films with different St/PL mass ratios. (**a**) TGA. (**b**) DTG.

**Figure 6 materials-15-02327-f006:**
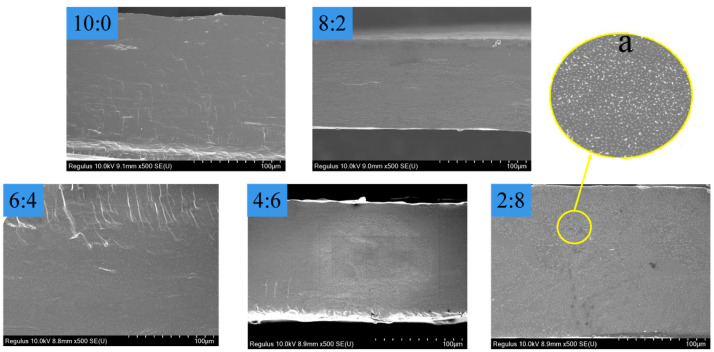
The SEM images of the composite films with different mass St/PL ratios. (a) Magnification ×10.

**Figure 7 materials-15-02327-f007:**
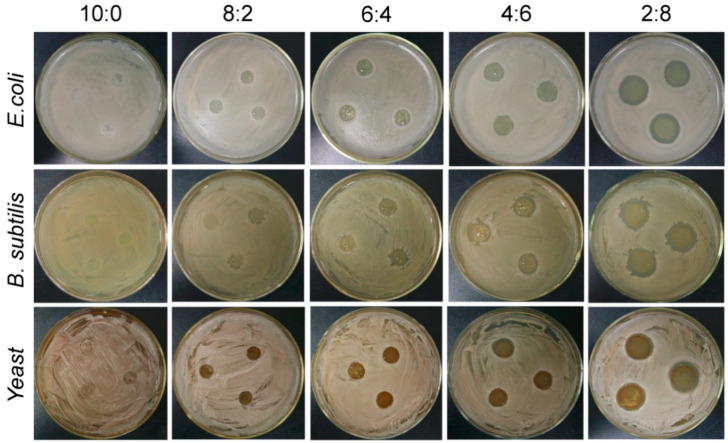
Antimicrobial activities of the composite films with different St/PL mass ratios with two bacteria and a fungus.

**Table 1 materials-15-02327-t001:** Diameter of the antimicrobial zones on the films exposed to three microbes.

	Diameter of the Antibacterial Zone (cm)				
	10:0	8:2	6:4	4:6	2:8
*E. coli*	0	1.00	1.18 ± 0.07	1.40 ± 0.10	1.98 ± 0.18
*B. subtilis*	0	1.2 ± 0.10	1.67 ± 0.07	1.83 ± 0.13	2.33 ± 0.13
Yeast	0	1.00	1.28 ± 0.05	1.45 ± 0.15	2.18 ± 0.08

## Data Availability

All data are freely available.

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
