# Peer review of "Preparation and Properties of Pea Starch/ε-Polylysine Composite Films"

_materials, 2022, doi:10.3390/ma15062327_

Round 1
Reviewer 1 Report
Manuscript “Preparation and Properties of Pea Starch/ε-Polylysine Compo- 2 site Films” is a very interesting study, clearly defined with appropriate reference to the literature. However, in my opinion, few questions need to be addressed:
- Methods, L 129: Please specify what are the species of used bacteria ( coli and S. aureus) and yeast.
- Methods, L 138: Please specify what incubation temperature was used for this assay.
- Results and Discussion, Figure 6, L 226: Magnification used for SEM pictures should be clearly stated.
- Results and Discussion, L 230-231: Antimicrobial effect of PL itself should be discussed, not just referenced.
- Conclusion might be improved, in a way to show the best outcomes of this study. Also, I would recommend discussion of results regarding packaging application in this section (previously mentioned in the Introduction).
Author Response
Manuscript “Preparation and Properties of Pea Starch/ε-Polylysine Compo- 2 site Films” is a very interesting study, clearly defined with appropriate reference to the literature. However, in my opinion, few questions need to be addressed:
Methods, L 129: Please specify what are the species of used bacteria ( coli and S. aureus) and yeast.
Response: We have added the relevant information accordingly.
Methods, L 138: Please specify what incubation temperature was used for this assay.
Response: We have specified this information in the revised manuscript.
Results and Discussion, Figure 6, L 226: Magnification used for SEM pictures should be clearly stated.
Response: We have added this detail accordingly.
Results and Discussion, L 230-231: Antimicrobial effect of PL itself should be discussed, not just referenced.
Response: We have discussed the antimicrobial effects of PL in this section.
Conclusion might be improved, in a way to show the best outcomes of this study. Also, I would recommend discussion of results regarding packaging application in this section (previously mentioned in the Introduction).
Response: We have revised the Conclusion section accordingly. As you pointed out, the packaging application has been mentioned in the Introduction section; therefore, we have not repeated it. I did the starch/polylysine film application work published in a Chinese journal.
Reviewer 2 Report
After reviewing the manuscript, I recommend it for publication in the Materials, due to the innovative nature of the presented results. However, I believe that the manuscript needs some corrections, which I listed below.
The characteristics and source of all materials used in experiments (also plasticizers) should be mentioned.
Materials and Methods:
Line 62: Were sodium alginate and glycerol added together for each sample? What was the expectation of such action?
Line 63: I suggest using the word dispersed instead of “dissolved”. Starch is insoluble in water.
Line 108: How were the tubes sealed? How was the film attached? “Salad oil” is not precise enough (whether its origin and properties do not affect the test results?).
Results and discussion.
Line 174: The last mentioned sample was probably confused. Should it not be the ratio” 8:2” at the end?
Line 159: Ts not TS, consequently.
Line 164-165: “Although the amount of PL increased due to its lower molecular weight…” This sentence is not clear. How could a lower molecular weight of PL influence its amount?
Line 169-170: The authors confirmed the PL role as a plasticizer. Therefore it raises curiosity why it was not possible to obtain a film with additional plasticizers (glycerol and sodium alginate). What could have happened in such an arrangement? Have the authors tried to explain this phenomenon?
FTIR analysis could be more precise and less superficial. In this form, it adds nothing interesting to the manuscript.
Line 185-186: How was the hydrogen bonding verified?
Line 196: “Meanwhile, the oil permeability of the composite films decreased with the increase of the proportion of PL” – decreased or increased? According to Fig. 4, oil permeability increased.
The conclusion should be rewritten to be more specific and decisive. They cannot be a shortened description of the obtained results using terms such as increased/ decreased or “changed regularly”.
Author Response
After reviewing the manuscript, I recommend it for publication in the Materials, due to the innovative nature of the presented results. However, I believe that the manuscript needs some corrections, which I listed below.
The characteristics and source of all materials used in experiments (also plasticizers) should be mentioned.
Response: We have added these details.
Materials and Methods:
Line 62: Were sodium alginate and glycerol added together for each sample? What was the expectation of such action?
Response: Yes, both sodium alginate and glycerol were added in each sample as both are plasticizers.
Line 63: I suggest using the word dispersed instead of “dissolved”. Starch is insoluble in water.
Response: We have used the term “dispersed” instead of “dissolved.”
Line 108: How were the tubes sealed? How was the film attached? “Salad oil” is not precise enough (whether its origin and properties do not affect the test results?).
We have added the following sentence to clarify these questions:
“Molten paraffin was dropped into the connection between the wall of the tube and the film, which formed a seal after cooling.”
We replaced “salad oil” with “peanut oil.” We believe the type of oil does not affect the test results.
Results and discussion.
Line 174: The last mentioned sample was probably confused. Should it not be the ratio” 8:2” at the end?
Response: The ratio is 2:8 at the end. At this ratio, a film can be prepared. We aimed to use the ratio St/PL (0:10), but a film could not be prepared at this ratio.
Line 159: Ts not TS, consequently.
Response: We have revised it accordingly.
Line 164-165: “Although the amount of PL increased due to its lower molecular weight…” This sentence is not clear. How could a lower molecular weight of PL influence its amount?
Response: Since a lower molecular weight PL cannot form a complex molecular network structure, the Ts is declined. We have revised this sentence for better clarity.
Line 169-170: The authors confirmed the PL role as a plasticizer. Therefore it raises curiosity why it was not possible to obtain a film with additional plasticizers (glycerol and sodium alginate). What could have happened in such an arrangement? Have the authors tried to explain this phenomenon?
Response: We think that PL can function as a plasticizer, but its molecular weight is more than that of a conventional plasticizer. The film of St had glycerol and sodium alginate in the ratio of 10:0, and the film of PL had glycerol and sodium alginate in the ratio of 0:10, which cannot be a film.
FTIR analysis could be more precise and less superficial. In this form, it adds nothing interesting to the manuscript.
Response: We agree with this comment. We used FTIR analysis only to illustrate that the two matrices have hydrogen bonding.
Line 185-186: How was the hydrogen bonding verified?
Response: We can see the vibration of the band at 1000~1750 cm-1 to redshift.
Line 196: “Meanwhile, the oil permeability of the composite films decreased with the increase of the proportion of PL” – decreased or increased? According to Fig. 4, oil permeability increased.
Response: We apologize for the error. Oil permeability increased. We have revised it accordingly.
The conclusion should be rewritten to be more specific and decisive. They cannot be a shortened description of the obtained results using terms such as increased/ decreased or “changed regularly”.
Response: We have revised the Conclusion section accordingly.